

# An Instrument for Direct Measurement of Emissions: Cooling Tower Example

Christopher D. Wallis[1], Mason D. Leandro[2], Patrick Y. Chuang[2], Anthony S. Wexler[1,3]

[1]Air Quality Research Center, University of California, Davis, CA, 95616, USA
[2]Department of Earth and Planetary Sciences, University of California, Santa Cruz, CA, 95064, USA
3Departments of Mechanical and Aerospace Engineering, Civil and Environmental Engineering, and Land, Air, and Water Resources, University of California, Davis, CA, 95616, USA

*Correspondence to*: Christopher Wallis (cdwallis@ucdavis.edu)



**Abstract.** Measuring emissions from stacks is challenging due to accessibility and safety concerns, and requires techniques to address a broad range of conditions and measurement challenges. One way to facilitate such measurements is to build an instrument package and then use a crane to hold the package over the emissions source. Here we describe such an instrument package that is used to characterize both wet droplet and dried

aerosol emissions from cooling tower spray drift. In this application, the instrument package characterizes the velocity, size distribution and concentration of the wet droplet emissions and the mass concentration and elemental composition of the dried $PM_{2.5}$ and $PM_{10}$ emissions. Subsequent papers will present and analyze the wet and dried emissions from individual towers.

## 1 Introduction

Cooling towers are used in a wide range of applications to dissipate waste heat to the environment. Wet cooling towers rely on the interaction between ambient air and cooling water to remove waste heat through evaporation. The interface between air and water results in some emission of liquid droplets, termed "spray drift." Drift emissions result in the release of aerosolized materials, which may result in undesirable consequences in terms of respirable particle emissions, mineral deposition in nearby areas, and biological concerns such as the spread of legionella

(Golay et al., 1986; Lucas et al., 2012; Mouchtouri et al., 2010). The U.S. Environmental Protection Agency (EPA) AP42 provides guidance on drift emissions based on measurements performed in the 1980s and 1990s. AP42 acknowledges a conservative calculation of emissions, and considers all emissions as $PM_{10}$, citing a lack of clear methodologies for accurately measuring both wet and dry tower emissions and characterizing the $PM_{2.5}$ fraction (EPA-AP42, 1995, p. 42).

A number of methods have previously been employed to characterize drift emissions. Comprehensive reviews of available characterization methods have found that while some methods do well in certain regards and under certain circumstances, no definitive method is suited to all ranges of conditions (Golay et al., 1986; Kinsey, 1991). Broadly, these methods may be categorized into those that measure elemental flux and those that measure droplet size distribution (Kinsey, 1991). Liquid water flux can be measured using thermodynamic methods employing

calorimeters or heated psychrometers to measure total liquid water output. Several methods rely on collection of spray drift residue using impingers or cyclone separators to provide an insight into overall drift mass flux by comparing measured mineral flux against tower water composition. Drift emissions have also been characterized using addition of tracer chemical markers to the recirculating water reservoir and monitoring the tracer over time (Campbell, 1969; Lucas et al., 2012). Such methods allow determination of total drift while avoiding the issue of

differentiating tower emissions from ambient particles. In general, mineral flux methods excel in high water emission





conditions, but exhibit poor performance and higher uncertainty with the lower water emissions that are increasingly common with the use of modern drift eliminators (Golay et al., 1986). These methods have varying degrees of effectiveness depending on droplet size, require collection of significant material per sample, and do not preserve information on drift droplet size distribution, which is necessary to determine particle transport and deposition (Golay et al., 1986; Kinsey, 1991; Roffman and Van Vleck, 1974).

A second class of measurements characterize emissions by counting spray drift droplets and using water composition data to predict total drift emission as well as dried aerosol emission diameters. Sensitive paper methods to detect droplet impactions have been used for many years and have benefitted from advances in digital image processing to increase throughput (Ruiz et al., 2013), but require high sampling numbers to achieve sufficient statistics and are limited to short sampling times to prevent saturation of the sensitive surface. Sensitive surface methods also require droplet impaction to collect samples, necessitating disruption of the aerosol stream (Golay et al., 1986). Microphotography and laser scattering techniques have been used to determine droplet size distribution optically and have potential to provide time resolution but are subject to interference from droplet coincidence at high concentrations and have accuracy and droplet size limitations (Kinsey, 1991). In general, droplet counting methods excel in lower water loading scenarios (Golay et al., 1986). All counting methods require large amounts of data collection to achieve satisfactory statistics, and may be prone to error for droplet sizes that are not as abundant. While these techniques provide valuable information regarding droplet number concentration and size distribution, small errors in measurement of droplet diameter can result in large errors in calculation of total drift mass flux (Kinsey, 1991). Additionally, droplet-based methods are largely unable to distinguish between spray drift and condensation formed as saturated air exits the tower (Kinsey, 1991; Ruiz et al., 2013).

In order to directly measure cooling tower emissions, we constructed an instrument package and suspended it directly above active cooling towers. Direct measurement of tower emissions allows emission characterization and collection while minimizing the effect of dilution, the difficulties with plume tracking and the need for data extrapolation. Subsequent papers will report the results of these measurements at individual power plant cooling towers.

## 2 Methods

### 2.1 System Components

A chassis was constructed of aluminum channel extrusion, built as an open frame to minimize disruption of air flow from the cooling tower exit. A sampling region at one end of the instrument chassis contains the aerosol sampling inlets, a phase Doppler interferometer (PDI), an updraft anemometer, and a temperature and humidity probe. Dried spray drift is sampled by drawing emissions through drying columns and subsequently distributing it to instrumentation for real-time size analysis as well as to filter-based PM samplers. Emissions are sampled by lifting



the instrument package with a crane and suspending it ~1 m over the top of the tower stack. Figure 1 shows the aerosol sampling train and sensors in the implementation used to characterize tower samples. Figure 2 shows the instrument package suspended above a cooling tower for sampling.

1. Updraft Anemometer
2. Phase Doppler Interferometer
3. Inlet Nozzles
4. Nafion Dryers
5. Plume Temperature and Humidity Probe
6. Dried Aerosol Manifold
7. Aerodynamic Particle Sizer
8. Dusttrak
9. IMPROVE $PM_{10}$ Sampler
10. IMPROVE $PM_{2.5}$ Sampler
11. IMPROVE Controller
12. IMPROVE Pumps
13. Dried Aerosol RH Probe
14. HEPA Filters
15. Flow Control Pump
16. Flow Meter
17. PDI Electronics
18. Electronics Enclosure

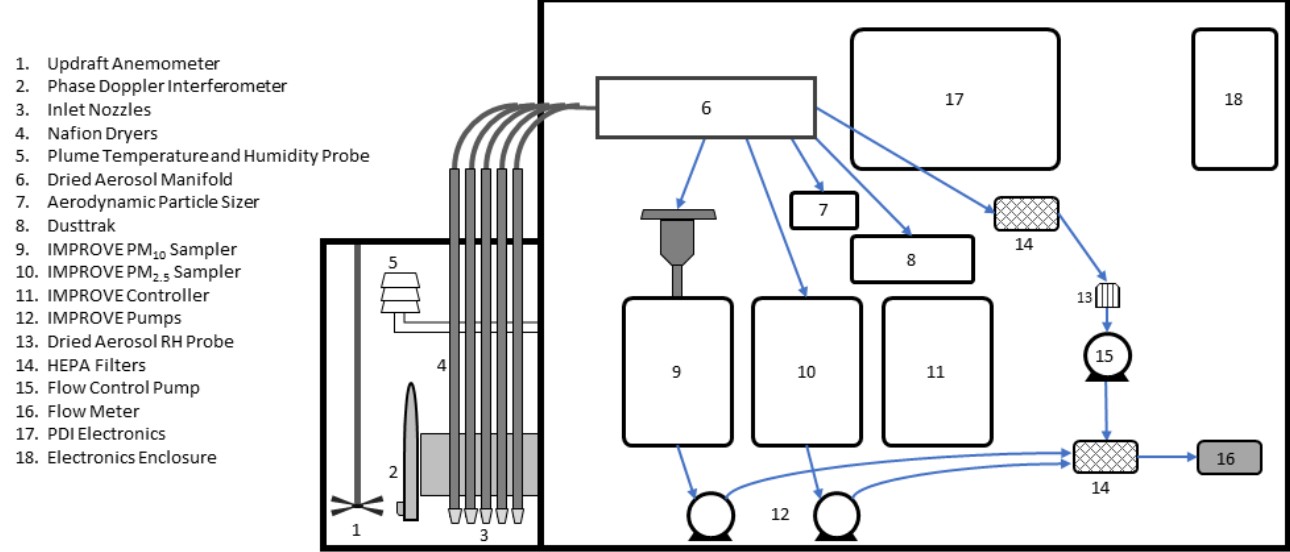

Figure 1. Sampling Instrumentation package a) diagram, b) right side, c) sampling end, d) left side



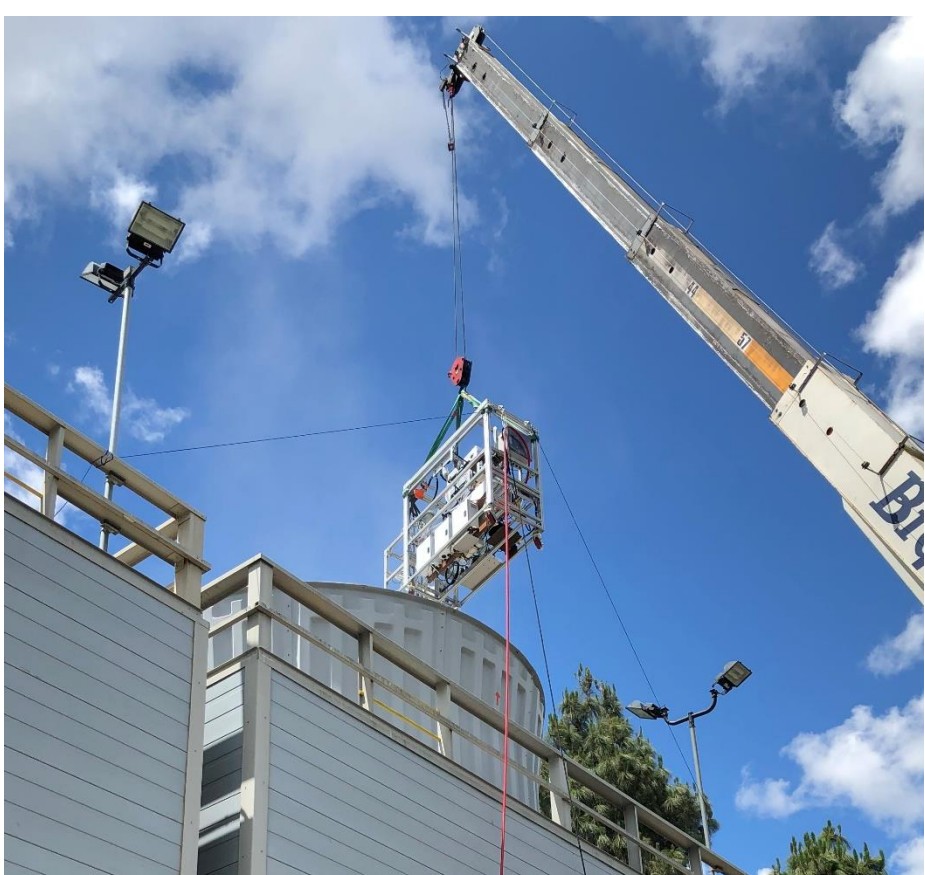

**Figure 2. Instrument package in place above a cooling tower**

### 2.1.1 Liquid Droplet Characterization

A phase Doppler interferometer flight probe (Artium Technologies Inc, Sunnyvale, CA) was mounted at the sampling end of the chassis to count and size liquid droplets emitted from the tower (Figure 1, item 2). Droplets passing through the PDI beam intersection result in phase shifts between multiple detector pairs which are used to precisely determine droplet velocity and diameter. This dual-range PDI measures drops in the range of 2 - 2000 μm (diameter) and provides droplet size and velocity (Bachalo, 2000). To derive population-level statistics such as number concentration and liquid water content, we use the method described by (Chuang et al., 2008). The PDI measurements are subject to a number of sources of uncertainty. The first is the uncertainty of the inferred diameter of any individual droplet, which is estimated from laboratory calibrations to be less than 1 μm. The second is uncertainty in the probe volume, which determines the sampling rate of the instrument in units of volume of air per unit time. We estimate the uncertainty to be about 5 to 10% (see Chuang et al. 2008 for details), with lower values at large drop sizes, and higher values at smaller drop sizes. The third is statistical counting uncertainty, which arises because the number of drops detected in any given time interval is subject to randomness. We estimate the





magnitude of this uncertainty using the well-known formula: for n droplets counted, then the uncertainty in this
count is $\sqrt{n}$. If the number of drops counted is more than $10^2$, then the relative uncertainty is less than 10%. PDI
electronics are housed in a protective enclosure covered in reflective material to reject solar heat. A low flow of
spent sheath air from the drying system is used to remove moisture from the enclosure.

### 2.1.2 Dried Aerosol Characterization

Dried aerosol is drawn through dryers and then distributed to various sampling instruments. Tower updraft velocity
is monitored by a Model 21706T Updraft Propeller Anemometer (R.M. Young Company, Traverse City, MI)
mounted at the sampling end of the instrument at the same height as the PDI and dryer inlet nozzles (Figure 1,
item 1). A variable speed oil-less piston pump (Figure 1, item 15) is used to control total instrument flow in order to
match the vertical component of the spray drift as it is drawn into the Nafion dryers (Figure 1, item 4). Total volume
flow in the system is calculated by combining and filtering the exhaust of the three main pumps driving flow in the
sampling system (IMPROVE $PM_{2.5}$, IMPROVE $PM_{10,}$ and variable speed flow control pump) and measuring the
combined volume flow using a mass flow meter (model 4100, TSI Incorporated, Shoreview, MN) (Figure 1, item
16). Static values of 5 lpm and 1 lpm respectively are added to account for the APS and Dusttrak flow rates. Volume
flow rate is converted to inlet nozzle velocity based on the diameter of the inlet nozzles installed. The feedback
loop for matching the vertical component of updraft velocity while sampling is shown in Figure 3. An overpressure
relief valve was included in the exhaust line to prevent accidental over pressurization of the flow meter.

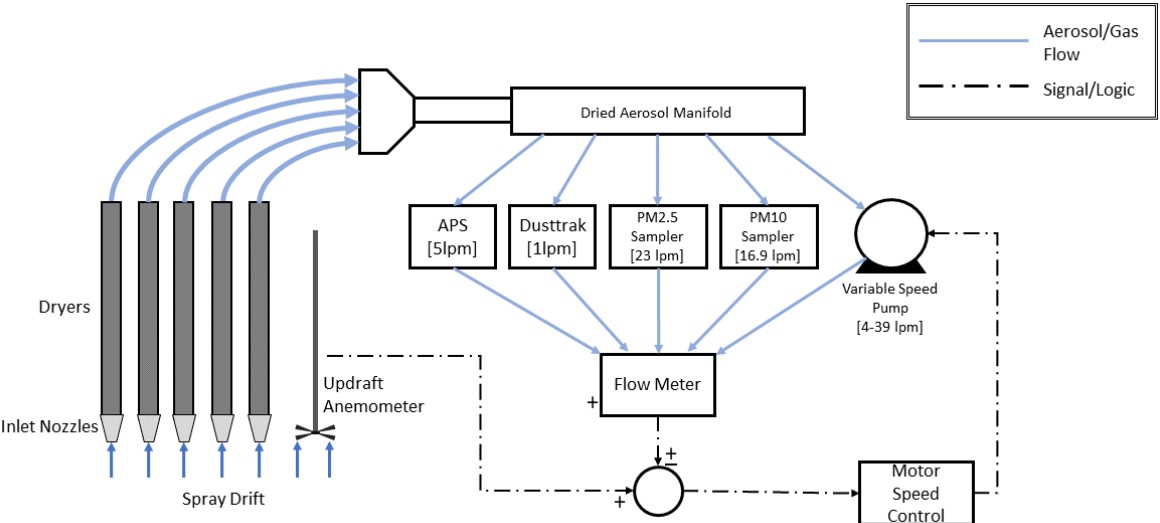

**Figure 3. Feedback control for matching vertical updraft velocity**


A feedback algorithm was implemented using an on-board microcontroller to minimize error between calculated
nozzle inlet velocity and measured updraft velocity by changing the speed of the flow control pump. Flow rates of
between 50-85 lpm are achievable with this method. Installation of inlet nozzles of different diameters allows an
adjustment to the range of inlet velocities that corresponded to this flow range (Table 1).

| NOZZLE DIAMETER, MM | ACHIEVABLE UPDRAFT VELOCITY, M/S |
|---|---|
| 4.2 | 12.0 - 20.4 |
| 4.9 | 9.0 - 15.3 |
| 5.5 | 7.0 - 11.9 |
| 5.9 | 6.0 - 10.2 |
| 7.3 | 4.0 - 6.8 |

**Table 1. Inlet nozzle diameters vs. nozzle velocity range**

In order to characterize dry emissions while sampling at the tower exit, spray drift must be dried while preserving
particle suspension. To emulate natural ambient drying, wet tower emissions are drawn in through a bank of Nafion
tubing dryers (model MD-700, Perma Pure, Toms River, NJ) located at the sampling end of the frame (Figure 1,
item 4). Each dryer consists of a central tube conducting sample aerosol surrounded by a counter-flow of dry
instrument-grade sheath air flowing at a minimum of twice the flow rate of the sample air. The two flows are
separated by a Nafion membrane, allowing diffusion of water from the sample air into the sheath air without heating
or diluting the aerosol sample. The bank of five 48-inch (122 cm) long dryers is operated in parallel to maintain an
aerosol flow rate less than 16.7 lpm per dryer, which is the design maximum flow rate for the MD-700. Dry sheath
air to each Nafion dryer is controlled using a variable area flowmeter. Figure 4 depicts a single MD-700 dryer
configuration.

A calculation of droplet drying capacity was performed using the method described by Hinds (Hinds, 1999). Initial
PDI testing detected maximum droplet sizes of approximately 100 μm exiting the tower. Relative humidity (RH)
downstream of the dryers was measured directly, and was approximately 60%. If the 1.2 m Nafion dryer tubes are
considered to have a linear humidity drop from saturation at the inlet to 60% RH at the outlet, the calculation can
be performed using an average of 80% RH in the dryer. At maximum design flow rate, aerosol has a residence
time of 5 seconds in each drying tube. This is sufficient time to completely dry droplets as large as 119 μm by the
time they exit the dryers. An additional 0.6 m of 19 mm conductive tubing downstream of the Nafion dryers has
additional drying potential.

Droplet shattering within the drying tubes was also considered, since this phenomenon would result in large drift
droplets being characterized as multiple small droplets. The ratio of the surface energy to the kinetic energy of
moving water drops was calculated for droplets from 1-100 μm and for velocities from 1-10 m/s (Pruppacher and


Klett, 2012). All ratios were at least 10x larger than the critical value at which substantial breakup occurs, meaning
droplet shattering is not expected.

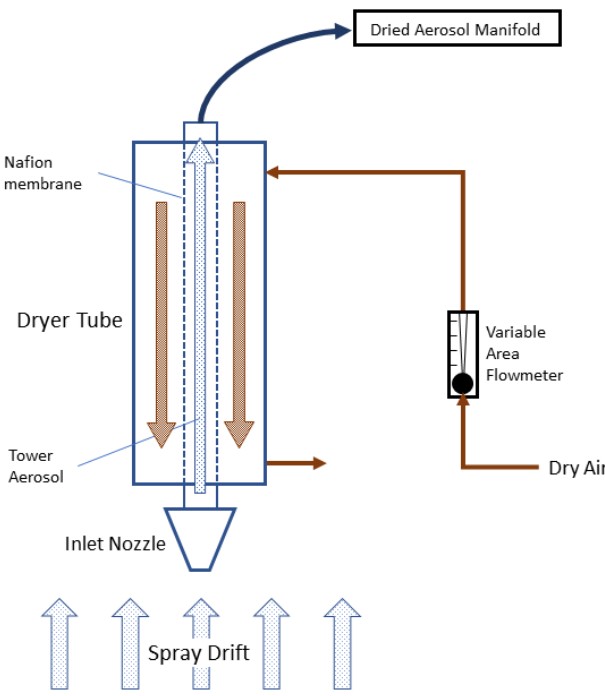

**Figure 4. Configuration for a single MD-700 dryer**

Dry filtered air for the MD-700 sheath is generated on-site using an oil-less piston compressor (model 7HDD-57-
M750X, Gast Manufacting, Inc, Benton Harbor, MI), an air-cooled aftercooler, and a refrigerated-type compressed
air dryer (model Krad-15, Keltec Technolab, Twinsburg, OH), generating approximately 170 lpm of 0.1 µm filtered
air with a dew point of -16C. Figure 5 shows the dry air generation system. The air is then regulated to 10 psig (~70
kPa gauge) and supplied to the MD-700 driers on the aloft instrument via a 1 inch (25 mm) diameter 150 ft (46 m)
umbilical hose. The hose is attached to the instrument chassis via a strain relief, and is also strain relieved at the
supply end near the dryer connection. The dry air is split into 5 separate flows for each of the Nafion dryer sheath
air paths via a bank of variable area flowmeters.





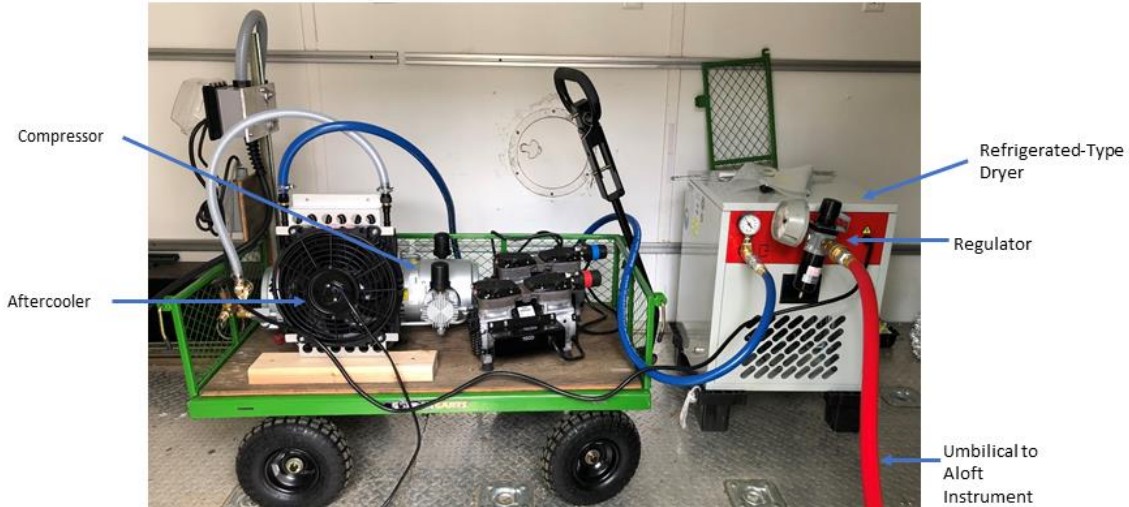

**Figure 5. Dry air generation equipment including compressor, aftercooler, dryer, regulator, and umbilical hose.**

Dried aerosol is then drawn to a manifold feeding a number of instruments (Figure 1, item 6). Grounded static dissipative tubing and lining are used to minimize particle loss due to accumulated surface charge. An aerodynamic particle sizer (APS) (Model 3321, TSI Incorporated, Shoreview, MN) is used for primary dry aerosol size and number quantification (Figure 1, item 7). A DustTrak (Model 8533, TSI Incorporated, Shoreview, MN) provides auxiliary characterization of particle size and concentration (Figure 1, item 8). A water-tight enclosure protects both instruments from liquid damage. A low-power notebook computer is used to log data and remotely begin and end sampling for both instruments.

Dried aerosol samples are also collected from the manifold using a pair of Interagency Monitoring of Protected Visual Environments (IMPROVE) sampler modules. A $PM_{2.5}$ module (Figure 1, item 10) collects sample at a rate of 23 lpm and a $PM_{10}$ module (Figure 1, item 9) with the sampling head modified for in-line flow collects sample at 16.9 lpm. Each sampler module contains four filter positions loaded with pre-weighed 25mm PTFE filters (Pall Teflo 3 μm, Pall Corporation, Port Washington, NY). Particulate matter gathered on these filters is subsequently analyzed gravimetrically and by X-ray fluorescence (XRF) to determine mass and elemental composition (IMPROVE, 2020a, 2020b, 2017). An IMPROVE control module (Figure 1, item 11) connected to both modules monitors sampling flow rates and is used to remotely start and stop the samplers over a cellular data connection.

## 2.2 Data Logging and Telemetry

Vital metrics are recorded using a custom electronics package based on a ruggedized version of an Arduino microcontroller (Rugged Mega, Rugged Circuits, MI) mounted on the aloft instrument (Figure 1, item 18). Updraft





velocity, aerosol flow rate, dried aerosol temperature and humidity, and tower plume temperature and humidity are recorded to a local SD card and transmitted to a computer on the ground in real time. A 2.4 GHz Zigbee wireless connection is used to transmit data to a ground computer and to receive commands from the ground to start or stop sampling and adjust pump parameters. Custom code developed in LabView receives, parses, and logs data from the aloft instrument, and allows remote control of flow and sampling state.

A local positioning system (model MDEK1001, Decawave Limited) is used to assist with precise and repeatable positioning of the aloft instrument rack. Four fixed anchor nodes are placed on the top deck of the cooling tower. A fifth sensor is mounted to the sampling end of the instrument, and relayed position data via serial output. Scale markers were also painted on the instrument chassis at 12-inch intervals for visual reference relative to tower features. A wireless camera (Casacam VS1001) was installed above the sampling end of the instrument to provide

a direct view of the sampler's position relative to the tower. Once the instrument is positioned in the desired location, two guy ropes are used to secure the instrument package to the tower structure to prevent unwanted motion due to air currents acting on the assembly. Once tethered, the instrument package is constrained to approximately 0.3 m of travel.

Tower plume temperature and humidity are continuously monitored by a probe mounted in the sampling end of the

chassis (model HMS112, Vaisala Inc, Louisville, CO). Dried aerosol temperature and humidity is monitored by an additional probe (model 657C-1, Dwyer Instruments, Michigan City, IN) (Figure 1, item 13) placed downstream of a HEPA filter (model HC10-4N-PTF, Aerocolloid LLC, Minneapolis, MN) (Figure 1, item 14) to protect the probe from contaminants. The humidity probe is positioned between the dry aerosol manifold and the flow control pump so that the probe does not affect aerosol being sampled by other instrumentation.

**2.3 Environmental Considerations**

The instrument package is required to operate in a variety of industrial environments, including electrically noisy environments, operation in heat and direct sunlight, and exposure to a constant upward flow of high humidity and water spray. To minimize potential impacts of electrical noise, analog sensor readings are transmitted as 4-20 ma current loop signals where possible. These signals are converted to digital and subsequently recorded by the

microcontroller.

Sensitive instrumentation and electrical connections throughout the instrument package are housed in watertight enclosures. Where additional cooling is necessary, spent dry air from the Nafion dryer sheaths is directed to electrical enclosures to cool instrumentation before exhausting to the environment. Exhaust air from the motor control pump is expelled through a row of small air jets, positioned to deflect and remove droplets from the camera

lens in order to maintain visibility. Instrumentation and connections that do not require additional cooling air are enclosed with desiccant packs as an additional precaution. Wireless temperature and humidity probes (Thermopro





TP65), are placed in electronics enclosures to provide real time indication of equipment conditions to operators on the ground.

## 2.4 Ambient Sampling

To differentiate between spray-drift-based emissions and ambient aerosol passing through the tower, a parallel set of measurements are made adjacent to the tower. An instrument package including an APS, Dusttrak, IMPROVE $PM_{2.5}$ and $PM_{10}$ samplers is operated concurrently with aloft sampling. Sampling inlets at the ambient sampling station are located at a height of 2 meters. Ambient temperature and humidity are continuously recorded using a model HL-1D data logger (Rotronic AG, Bassersdorf, Switzerland). Wind speed and direction near the tower are logged to a local SD card on a microcontroller attached to a battery-powered sonic anemometer (Model 81000, RM Young Company, Traverse City, MI).

## 2.5 Power

Power for operating ground and aloft instruments is generated on-site using a 1000 W propane-powered generator. A dual-tank switching regulator allows the propane tanks to be exchanged without interrupting sampling. Generator exhaust is cooled via dilution with ambient air drawn in through an in-line blower (model FR110, Fantech, Lenexa, KS) and then HEPA filtered (model CFB-HP-6, HVACQuick, Medford, OR) prior to release to prevent contamination of sample aerosol at the site. A temperature probe is mounted in the dilute exhaust stream to ensure that exhaust gases do not damage the filter. The generator is positioned approximately 100 feet (30 m) from the ground sampling station and cooling cell, as shown in Figure 6. Power is delivered to the ground and aloft instruments via a 150 ft (46 m) supply umbilical, strain relieved near either end to the instrument chassis and to a stationary block near the generator.



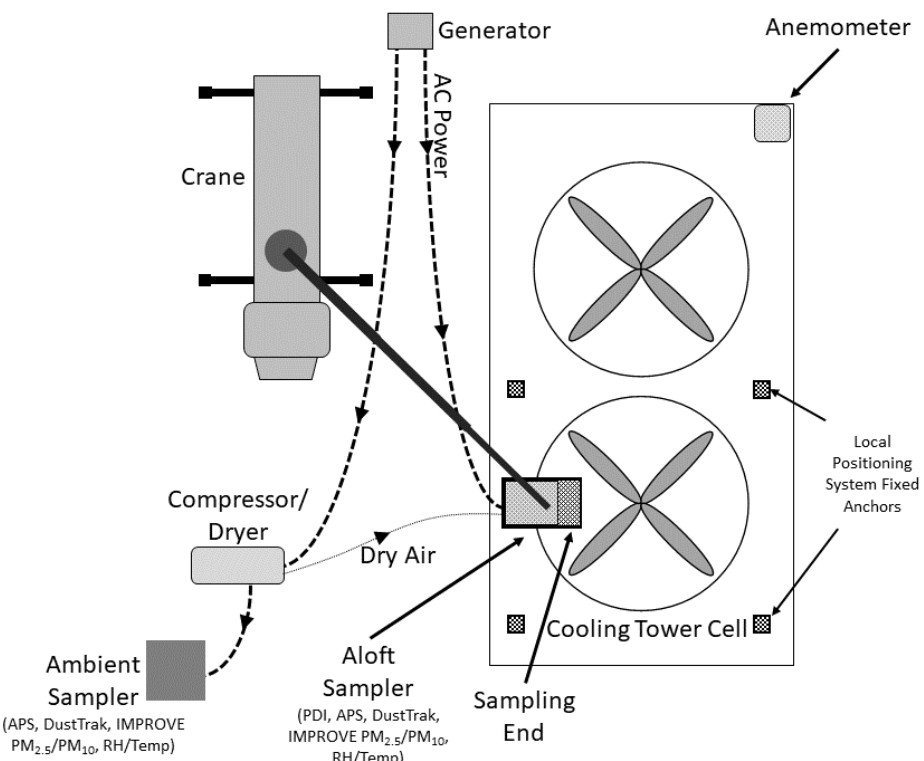

**Figure 6. Typical deployment for tower sampling (view is from above)**


## 2.6 Additional Considerations

Due to the nature of the sampling method, it is critical that no equipment or components could fall from the crane or instrument chassis due to the risk of damage to the plant facility. All equipment on the instrument platform is redundantly secured. Components that can potentially fall if one or two fasteners fail are secured with tether cables

as an additional precaution.

The complete aloft instrument weighs approximately 800 lbs (~360 kg). It is fitted with shock-absorbing locking casters and can be moved on level ground by one or two people. All supplemental field equipment, including sampling equipment, compressors, dryers, generators, and exhaust filtration is mounted on carts or otherwise easily portable for field deployment.

## 2.7 Sampling Validation

Aloft and ground IMPROVE samplers and APS's were run in parallel in a collocated study within a well-mixed room to determine sampling differences. Gravimetric results are shown in Table 2. The discrepancy in collection of larger





PM is presumed to be due to losses within the aloft sampling train. APS instrument variability was observed both with the entire aloft sampling train in place and with bare APS instruments placed side-by-side. To account for this
variation, collocated samples were taken before and after each field sampling day, in which the two APS units were allowed to run with the aloft instrument placed on the ground near the ground sampling station. Data from the aloft APS was normalized to the ground APS for each site based on this collocation period. Normalization was applied for size ranges below 12 μm. A discrepancy was seen for particles larger than 12 μm that were likely lost to impaction on the way to the APS. Therefore, data for this size range is not reported. In addition, aloft APS data was
normalized to ground data to account for line losses leading to the aloft APS, which were characterized during the extended ground collocation test.

| TEST 1 | PM2.5 [UG/M3] | PM10 [UG/M3] | COARSE (PM10 - PM2.5) [UG/M3] |
|---|---|---|---|
| **ALOFT INSTRUMENT** | 8.93 | 11.41 | 2.48 |
| **GROUND INSTRUMENT** | 8.93 | 12.01 | 3.09 |
| | | | |
| **TEST 2** | PM2.5 [ug/m3] | PM10 [ug/m3] | Coarse (PM10 - PM2.5) [ug/m3] |
| **ALOFT INSTRUMENT** | 12.24 | 15.54 | 3.30 |
| **GROUND INSTRUMENT** | 12.22 | 16.14 | 3.92 |

**Table 2. Gravimetric results from IMPROVE sampler collocation. PM coarse is calculated as the difference between PM10 and PM2.5**

**3 Discussion**

A number of previous methods have been used to characterize cooling tower emissions. Ultimately, tower emissions characterization is concerned with measuring the aerosol that leaves the facility and enters the environment or settles to the ground – in the case of wet cooling towers, this takes the form of dried aerosol composed of the contents of spray drift droplets. However, all techniques used in measurement of cooling tower
emissions characterize liquid droplets escaping the tower rather than the eventual dry aerosol. Ambient and fence line monitoring techniques such as those applied to refinery emissions sample aerosol that is realistically aged and actually leaves the facility, but can only characterize small portions of overall emissions and cannot easily be related to total emissions without knowing how dilute the aerosol has become as it mixes with the environment. Attempts to correct for ground sampling dilution have resulted in order-of-magnitude discrepancies (Roffman and Van Vleck,



1974). Direct sampling above the tower in conjunction with use of an on-board aerosol drying system allows direct characterization of dried aerosol, without unknown levels of dilution from mixing with ambient air. Rapid drying of spray drift mimics environmental drying while preserving dry aerosol size distribution, in contrast to bulk droplet collection methods such as heated glass bead sampling. This size preservation allows size-segregated sampling of dried aerosol into coarse and fine ranges for collection and subsequent analysis with IMPROVE $PM_{2.5}$ and $PM_{10}$

modules. Pre-drying also enabled detailed dried aerosol size characterization using real-time instrumentation such as the APS, which provides size characterization between 0.5 µm to 12 µm. This combination allows direct characterization of aerosol that ultimately is emitted to the environment, and eliminates false signal from water droplets formed by re-condensation of saturated air exiting the tower rather than spray drift. In contrast, existing methods that characterize emissions based on droplet counts, including sensitive surface and optical methods, are

all subject to this issue. The APS completes each sample analysis in approximately one minute, while longer sampling times are required for bulk sample collection using IMPROVE samplers and filters to collect sufficient mass for analysis.

In conjunction with multiple characterizations of dried aerosol, drift droplets are characterized with a PDI, which offers high accuracy, time resolved data. The PDI characterizes liquid droplets without disturbing the aerosol flow

as it exits the tower, and operates with less than 10% relative size uncertainty over a range of 2 µm to 2mm. The PDI provides higher accuracy than previous light scattering methods, particularly in smaller droplet size ranges (Bachalo, 2000). These instruments provide rapid characterization, requiring as little as 5 minutes of sampling at a single location above the tower to obtain a statistically significant sample size.

Specific instrumentation chosen for the instrument package, including the PDI and APS, allow high resolution size

characterization compared to methods based on scattering, photography, or sensitive paper. Use of IMPROVE samplers for size-selected sample collection allows a distinction between coarse and fine particles when analyzing for total mass as well as composition. On-board rapid droplet drying preserves the aerosol size distribution enabling thorough characterization of dried emissions, while avoiding issues of false sampling of re-condensation that affect many other droplet sampling methods. Use of multiple sampling modalities (i.e. droplet, dried aerosol, optical, and

elemental analysis) provides a range of analytical strengths not available with the prior sampling techniques outlines above.

**4 Conclusion**

Spray drift emissions are difficult to accurately characterize for a number of reasons, and past attempts have not resulted in a definitive method. Direct sampling from the tower exit combined with rapid heatless drying enables

representative sampling while preserving aerosol properties such as size distribution. Inclusion of modern instrumentation in the sampler package allows enhanced measurement precision compared to past techniques,



and size-resolved collection of dried aerosol enables separate analysis of $PM_{2.5}$ and $PM_{10}$ mass emissions and chemical composition. The instrumentation described here is designed for characterization of wet cooling tower emissions, but is also suitable for measurement of droplet-, aerosol- or gas-based emissions from a variety of
sources. The instrument rack configured with different instruments could, in general, be used to measure emissions from any stack.

## Acknowledgement

This work was supported by California Energy Commission contract EPC-16-040.

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
