# Peer review of "An Instrument for Direct Measurement of Emissions: Cooling Tower Example"

_Atmospheric Measurement Techniques, 2021_

## Author Response (AR1)

**Reviewer 1 Comments:**

*[General comments]*

*Comment: This manuscript describes the development of an instrument package for characterizing cooling tower spray drift emissions of both wet droplets and dried aerosols. The instrument package measures the velocity, size distribution and concentration of the wet droplet emissions, as well as the mass concentration and elemental composition of the dried $PM_{2.5}$ and $PM_{10}$ emissions. Such a package provides increased measuring accuracy and is also suited for a range of measurements. The manuscript is well structured and easy to follow. However, a few questions should be addressed and the reviewer would appreciate further discussion in the manuscript. The reviewer recommends a minor revision of this manuscript before its publication.*

**Thank you for the feedback, we've found it to be very constructive. We have incorporated your suggestions into our revision as described below.**

*[Specific comments]*

*Comment: Line 87, please describe the methodology used by Chuang et al., 2008. It's difficult to locate it if being explained elsewhere.*

**We have expanded the description of the PDI methodology.**

*Comment: Section 2.1.2, Line 99 - 110, highlighting the item number for each item, some of which have, others of which do not, such as Anemometer, IMPROVE PM2.5, etc. Consistency assists readers in recognizing.*

**We have included references to all numbered items from Figure 1 in the revised manuscript text**

*Comment: Table 1, good information and summary, how are they calculated or are there any references cited?*

**Flow rates are calculated based on the dividing the available volume flow rate by the cross sectional area of 5 nozzles. The text has been updated to mention this.**

*Comment: Line 165, perhaps include the word "IMPROVE" in the first sentence stated in the text Line 105.*

**Thank you for the comment, we have corrected this.**

*Comment: Line 163, state the model of this low-power notebook computer that was utilized.*

**We have updated the text to include this.**

*[Technical corrections]*

*Comment: Line 212 "a parallel set of measurements are made adjacent to the tower" suggest changing "are" to "is"*

**We have updated the text to include this.**

*Comment: Line 213 "IMPROVE PM2.5 and PM10 samplers is operated…" suggest changing "is" to "are"*

**"Is" refers to "an instrument package" in this case, and so should remain singular.**

*Comment: Figure 1a, the font of item "13" is smaller than the rest of them. Keep consistency.*

**Thank you for the comment, we have corrected this.**

*Comment: References "IMPROVE, 2020a. IMPROVE, 2020b. IMPROVE, 2017." Pages are not found. The website turned out: "The IMPROVE website has recently been updated and reorganized, and many links from the old site are no longer active."*

**We have updated the reference to direct readers to the IMPROVE documentation page where updated versions of these documents will continue to be maintained.**

**Reviewer 2 Comments:**

*The manuscript presents a method to measure aerosol emission from cooling towers using direct dried-aerosol sampling techniques, both off- and online. The authors use two sets of instruments; aloft and ground-based. The latter one is used as reference to correct for sampling losses in the aloft setup. The normalization is done using PM10 and PM2.5 passive sampling but a comparison of the APS results is missing. I suggest the auhors to provide a comparison of the size distributions obtained by APS measurements in order to improve the manuscript. The paper is well written and should be published after considering this suggestions and the following minor comments.*

**Thank you for the comments and suggestions, we find them to be helpful. We will incorporate them into the revision of the paper. We will compare APS results from the two sampling trains along with field findings in a subsequent manuscript describing dried aerosol findings.**

*Minor Comments:*

*What would be the expanded uncertainty of the method?*

**We have updated the discussion of this method to include total uncertainty.**

*How far from the tower is the parallel set of measurements located?*

**The exact distance from the tower to the parallel measurement varied between towers, but was approximately 100 feet (30m). Positioning of the ground-based instrument was such that it was representative of the intake of the cooling tower but avoided emissions from the ground-based generator and exhaust from the crane, and was subject to different physical layout constraints at each site. We have updated the manuscript to include this information.**

*Table 2: Please report standard deviations for the values shown here. Please fix the units to µg/m³*

**The chart shows two individual test results, so there is no standard deviation to report. We have added uncertainty for this data collection method to the table.**